# Cytogenetic characterization and mapping of the repetitive DNAs in *Cycloramphus bolitoglossus* (Werner, 1897): More clues for the chromosome evolution in the genus *Cycloramphus* (Anura, Cycloramphidae)

Gislayne de Paula Bueno[1], Kaleb Pretto Gatto[2], Camilla Borges Gazolla[1], Peterson T. Leivas[3], Michelle M. Struett[4], Maurício Moura[4], Daniel Pacheco Bruschi[1]*

1 Departamento de Genética, Setor de Ciências Biológicas, Universidade Federal do Paraná (UFPR), Curitiba, Paraná, Brazil, 2 Departamento de Biodiversidade e Centro de Aquicultura, Instituto de Biociências, Universidade Estadual Paulista, (UNESP), Rio Claro, São Paulo, Brazil, 3 Curso de Ciências Biológicas, Universidade Positivo (UP), Curitiba, Paraná, Brazil, 4 Departamento de Zoologia, Setor de Ciências Biológicas, Universidade Federal do Paraná (UFPR), Curitiba, Paraná, Brazil

* danielpachecobruschi@gmail.com, danielbruschi@ufpr.br

## Abstract

*Cycloramphus bolitoglossus* (Werner, 1897) is a rare species with a low population density in the Serra do Mar region of Paraná and Santa Catarina, in southern Brazil. Currently, it has been assigned to the Near Threatened (NT) category in the Brazilian List of Endangered Animal Species. Here, we described the karyotype of this species for the first time and investigated the patterns of some repetitive DNA classes in the chromosomes using molecular cytogenetic approaches. We isolated, sequenced and mapped the 5S rDNA and the satellite DNA PcP190 of *C. bolitoglossus*, as well as mapped the telomeric sequences and seven microsatellites motifes [(GA)$_{15}$, (CA)$_{15}$, (GACA)$_4$, (GATA)$_8$, (CAG)$_{10}$, (CGC)$_{10}$, and (GAA)]$_{10}$. *Cycloramphus bolitoglossus* has 2n = 26 chromosomes and a fundamental number (FN) equal to 52, with a highly conserved karyotype compared to other genus members. Comparative cytogenetic under the phylogenetic context of genus allowed evolutionary interpretations of the morphological changes in the homologs of pairs 1, 3, and 6 along with the evolutionary history of *Cycloramphus*. Two subtypes of 5S rDNA type II were isolated in *C. bolitoglossus* genome, and several comparative analysis suggests mixed effects of concerted and birth-and-death evolution acting in this repetitive DNA. The 5S rDNA II subtype "a" and "b" was mapped on chromosome 1. However, their different position along chromosome 1 provide an excellent chromosome marker for future studies. PcP190 satellite DNA, already reported for species of the families Hylidae, Hylodidae, Leptodactylidae, and Odontophrynidae, is scattered throughout the *C. bolitoglossus* genome, and even non-heterochromatic regions showed hybridization signals using the PcP190 probe. Molecular analysis suggests that PcP190 satellite DNA exhibit a high-level of homogenization of this sequence in the genome of *C. bolitoglossus*. The PcP190 satDNA from *C. bolitoglossus* represents a novel sequence group, compared to other anurans, based on its hypervariable

**Data Availability Statement:** All relevant data are within the paper and its Supporting information files.

**Funding:** The author(s) received no specific funding for this work.

**Competing interests:** The authors have declared that no competing interests exist.

region. Overall, the present data on repetitive DNA sequences showed pseudogenization evidence and corroborated the hypothesis of the emergence of satDNA from rDNA 5S clusters. These two arguments that reinforced the importance of the birth-and-death evolutionary model to explain 5S rDNA patterns found in anuran genomes.

## Introduction

Frogs of the genus *Cycloramphus* Tschudi, 1838 are charismatic anurans restricted to the Brazilian Atlantic forest domain, with the most remarkable species richness associated with southern and southeastern highlands of Brazil. The genus is composed of 28 species [1], many of which have limited geographic ranges. *Cycloramphus bolitoglossus* (Werner, 1897) is a rare species with a low population density in the Serra do Mar region of Paraná and Santa Catarina, in southern Brazil. This poorly-known species is currently classified as Data Deficient (DD) by the International Union for Conservation of Nature [2, 3], although it has been assigned to the Near Threatened (NT) category in the Brazilian List of Endangered Animal Species [4].

Ten *Cycloramphus* species have been investigated cytogenetically, and most of the available data are limited to conventional karyotype descriptions using C-banding and silver impregnation [5–8]. The diploid number (2n = 26) is conserved in all the species, with karyotypes that are generally composed of metacentric and submetacentric pairs. Little variation has been identified in the fundamental number (FN), with FN = 50 in *C. boraceiensis* Heyer, 1983, *C. dubius* (Miranda-Ribeiro, 1920), *C eleutherodactylus* (Miranda-Ribeiro, 1920), and FN = 52 in *C. lutzorum* Heyer, 1983, *C. fuliginosus* Tschudi, 1838, *C. acagatan* Tschudi, 1838, *C. brasiliensis* (Steindachner, 1864), *C. carvalhoi* Heyer, 1983, *C. rhyakonastes* Heyer, 1983, *and C. asper* Werner, 1899 [5–8]. Pericentromeric inversions or translocations are the primary sources of karyotype variation in this group [5, 7]. The inclusion of new chromosomal markers with a phylogenetic approach to the analysis of the cytogenetic characteristics of the genus *Cycloramphus* [9] should contribute to the understanding of the mechanisms of chromosomal evolution that have molded the karyotypes of this genus.

Repetitive DNAs are excellent chromosomal markers for comparative genomics at the chromosome level [10]. The 5S rDNA is an *in tandem* repeated multigene family in the genome, and it has been widely used in chromosome studies [11]. It consists of a coding sequence of 120 base pairs (bps) followed by a non-transcribed spacer (NTS) of variable size and composition [11, 12]. The coding sequence contains an internal control region (ICR) composed of some elements characterized as box A, an intermediate element (IE), and box C, with some variation being found among the different taxa in which this multigene family has already been studied [11–13].

The highly conserved nature of the 5S rDNA found among distantly related taxa has been confirmed in innumerable studies. The concerted evolution model proposes that a modification in a nucleotide position can be spread from one repeat unit to another through a process of homogenization [11, 14]. However, evolutionary mechanisms related to the Birth-and-Death theory have also been reported in this multigene family [15–18]. This is due to changes in nucleotide sequences through insertions, deletions, association with transposable elements, and the amplification of microsatellites, which cause the loss of regulatory regions, leading to a pseudogenization process [18, 19]. A mixed model of Concerted and Birth-and-death evolution has also been proposed to explain the evolution of this multigene family [19–21]. In amphibians, these processes may account for the emergence of new types of repetitive DNA sequences, such as the PcP190 satellite DNA (satDNA) family [22].

The PcP190 is a satDNA derived from the 5S rDNA, and up to now, it has been found in the genomes of some anuran species of the families Hylidae, Hylodidae and Leptodactylidae, and Odontophrynidae [10, 22–27]. The presence of this satDNA in different evolutionary branches of the Amphibian tree of life has stimulated considerable interest in the evolutionary significance of this repetitive sequence in the genome [10].

Short repetitive DNAs, such as microsatellites repeats, are also spread among the vertebrate genome and provide important insights into chromosomal evolution. Microsatellite repeats correspond to monomeric units of 2–7 bps [28–30], with the head-to-tail organization of a large number of repetitions in a chromosomal cluster. The chromosomal mapping of these sequences has provided a useful marker for comparing karyotypes in chromosomal evolution [31–35].

Here, we provide the first description of the karyotype of *C. bolitoglossus* based on both conventional and molecular cytogenetic methods, and we extend the known occurrence of the PcP190 satDNA to the family Cycloramphidae. We analyze the cytogenetic data available for the genus *Cycloramphus* in the context of the phylogenetic relationships of the genus [9] to understand the chromosome shifts that have occurred during the evolutionary history of this group. We also discuss the interplay among the 5S rDNA, microsatellite motifs, the PcP190 satDNA, and the potential for applying these features as cytogenetic markers in the genus *Cycloramphus*.

## Materials and methods

### Biological samples and classical cytogenetic analysis

Four *C. bolitoglossus* specimens (three males and one female) were caught from Pico do Marumbi State Park (25˚29'23" S; 48˚58'37" W) in the municipality of Piraquara, Paraná state, southern Brazil. Specimen collection was authorized by the Brazilian Institute for the Environment and Natural Resources (IBAMA, Instituto Brasileiro do Meio Ambiente e Recursos Naturais: process number 10277–1), and the Paraná State Environment Institute (IAP/Instituto Ambiental do Paraná: process number 3716). The tissue samples were extracted from the specimens after they had been euthanized through the application of 5% Lidocaine to the skin, to minimize abnormal suffering, following the recommendations of the Herpetological Animal Care and Use Committee (HACC) of the American Society of Ichthyologists and Herpetologists (available at: http://www.asih.org/publications), and approved by SISBIO/Chico Mendes Institute for the Conservation of Biodiversity (collecting license number 10277–1). Voucher specimens (MHNCI 11013–11016) were deposited in the Capão da Imbuia Natural History Museum in Curitiba, Brazil.

Chromosome preparations were obtained from intestinal epithelial cell suspensions, according to King and Rofe [36] and Schmid [37]. The chromosomes were stained with 10% Giemsa and silver impregnation was based on the Ag-NOR method described by Howell and Black [38]. To define the heterochromatic pattern, the chromosomes were submitted to C-banding techinique, following Sumner [39]. To better visualize the heterochromatic blocks, the chromosomes were stained with a 0.4 µg/mL solution of DAPI fluorochrome (6-diamidino-2-phenylindole) following the C-banding. Ten metaphases of two individuals were analyzed with Drawid 0.26 software [40] and chromosomes were ordered and classified by their centromeric position using the nomenclature proposed by Green and Session [41].

### Isolation of the 5S rDNA and PcP190, cloning, and molecular characterization

The genomic DNA was extracted from samples of liver and muscle tissue of all specimens (MHNCI 11013–11016) using the TNES method, according to Bruschi et al. [42]. The integrity

and quality of the material were verified by electrophoresis and it was quantified using Nano-Drop™ 2000/2000c spectrophotometers (Thermo Fischer Scientific, Waltham, USA). The 5S rDNA was amplified using the Polymerase Chain Reaction (PCR) based on 30ng/uL of the DNA template, 1x PCR buffer (200 mM Tris-HCl (pH 8.4), 500 mM KCl), 1.5 mM of MgCl$_2$, 0.3 mM of dNTP, 5 U/μL of *Taq* polymerase, and 0.5 μM of each primer in a reaction with 15 μL. The 5S rDNA was isolated using the primers 5S-A (5′ -TACGCCCGATCTCGTCCGATC −3′) and 5S-B (5′−CAGGCTGGTATGGCCGTAAGC−3′) [43]. The PcP190 satDNA was isolated using primers P190F (5′ - AGACTGGCTGGGAATCCCAG−3′) and P190R (5′− AGCTGCTGCGATCTGACAAGG−3′), also with the reaction protocol described above [22]. The amplification program was 94˚C for 5 minutes, followed by 30 cycles of 94˚C for 1 minute; 56˚C for 5S rDNA and 60˚C for satDNA PcP190 for 50 seconds; 72˚C for 1 minute, and a final extension with 72˚C for 8 minutes. The amplicons were purified using Wizard® SV Gel and the PCR Clean-Up System kit (Promega Corporation, Madison, USA), according to the manufacturer's instructions. The amplicons were cloned using the CloneJET PCR Cloning kit (ThermoFisher Scientific, Waltham, USA), and the vectors were used to transform *Escherichia coli TOP10*. After plasmidial extraction [44], we used the clones for PCR amplification, using the primers of the vector kit (forward: 5′ - CGACTCACTATAGGGAGAGCGGC−3′ and reverse: 5′ - AAGAACATCGATTTTCCATGGCAG−3′).

The DNA was sequenced bidirectionally using the Big Dye Terminator kit (Applied Biosystems, Foster City, USA), according to the manufacture's recommendations and sequenced in an ABI/Prism automatic sequencer (Applied Biosystems, Foster City, USA). The sequences were inspected and edited in Bioedit software 7.2.5 [45].

The rDNA 5S and PcP190 nucleotide sequences obtained in the present study were compared with sequences of other anurans available in the GenBank (NCBI) database (for GenBank accession numbers, see S1 and S2 Tables). These sequences were aligned using the MUSCLE algorithm [46] in MEGA X [47], and the similarity analysis was run in MEGA X by pairwise distance computation using the p-distance method. Similarity comparisons were conducted among four groups of sequences. Among the four groups, here are comparisons using only sequences isolated from *C. bolitoglossus*. Other comparisons sequences recovered from *C. bolitoglossus* were compared with sequences of other anurans from GenBank. From this, the following groups were established: (a) only the 5S rDNA sequences recovered from *C. bolitoglossus*, (b) presumable 5S rDNA transcribed regions from *C. bolitoglossus* and all 5S rRNA gene sequences of anurans available on GenBank; (c) all the completely conserved regions of the PcP190 satDNA monomers of *C. bolitoglossus* and sequences already available on GenBank for this satDNA, and (d) the presumable 5S rDNA transcribed region with the conserved region of PcP190 satDNA of *C. bolitoglossus* and other anurans available on GenBank.

A Maximum Likelihood tree was estimating using RAxML version 8.0.0 [48] (with the Kimura-2-parameter model with the gamma distribution heterogeneity across sites to delimit the groups of sequences for pairwise comparisons to define the types and subtypes of rDNA 5S. All the sequences generated in the present study were deposited in GenBank (accession numbers: MT920589—MT920617).

## Fluorescent in situ hybridization assays (FISH)

The 5S rDNA clones CB5STIIa.C1 and CB5STIIb.C7 were labeled by PCR using digoxigenin 11-dUTP (Roche, Mannheim, Germany). The probes for PcP190 satDNA (clone C7M2.PcP) were labeled with Biotin-16-dUTP using the Biotin Nick Translation Mix (Roche, Mannheim, Germany). These probes were precipitated in the presence of sonicated salmon sperm DNA (100 ng/μL), with 3M sodium acetate and ethanol. The cocktail of probes used for the

resuspension was prepared with 50% of formamide, 10% of dextran sulfate, and 2x SSC. The probe solutions were applied to the slides with the chromosome preparations after denaturation at 95˚C for 10 minutes. The slides were incubated for 72 hours at 37˚C following the protocol described by Pinkel et al. [49]. The digoxigenin probe was detected using an anti-digoxigenin antibody conjugated with rhodamine (Roche, Mannheim, Germany). The biotin-labeled probes were detected using streptavidin conjugated with Cy3 (Invitrogen, Carlsbad, USA).

We mapped the microsatellite motifs $(GA)_{15}$, $(CA)_{15}$, $(GACA)_4$, $(GATA)_8$, $(CAG)_{10}$, $(CGC)_{10}$, and $(GAA)_{10}$ using oligonucleotide probes marked directly with Cy5- fluorochrome (Sigma-Aldrich, St. Louis, USA) at the 5' end during their synthesis. The telomeric $(TTAGGG)_n$ probes were produced by PCR amplification using the telomeric primers F (5' TTAGGG 3') and R (5' CCCTAA 3') marked directly by the incorporation of 11-digoxigenin-dUTP, following the protocol described by Guerra [50].

The FISH experiments with microsatellite probes followed the protocol described by Kubat et al. [51], while the telomeric sequences were mapped using the procedures described by Ernetti et al. [52]. The chromosome preparations were counterstained with DAPI (0.4μg/mL). The images were captured using a Nikon confocal A1R MP microscope and an Olympus BX51 fluorescence microscope, and edited using Adobe Photoshop CS4 only for brightness and contrast.

## Results

### Description of the karyotype and the chromosomal mapping of the microsatellites and telomeric repeats

*Cycloramphus bolitoglossus* has 2n = 26 chromosomes with a karyotype composed of eleven metacentric pairs (1, 4, 5, 6, 7, 8, 9, 10, 11, 12 and 13) and two submetacentric pairs (2 and 3) with FN = 52 (Fig 1A and S3 Table). A secondary constriction was observed in an interstitial region of the long arm of pair 6, coinciding with the NOR site detected by the Ag-NOR method (Fig 1B). Heterochromatin, using C-band+DAPI procedure, was observed in pericentromeric blocks in the homologs of pairs 1, 3, 4, and 7, while in pairs 1 and 4, constitutive heterochromatin blocks were observed in both chromosome arms (Fig 1C). Pairs 3 and 7 presented signals only in the long arms. Additional C-bands+DAPI were detected in an interstitial position in the long arms of the homologs of pairs 8 and 10 (Fig 1C). Negative DAPI staining was also observed in pair 6, coinciding with the secondary constriction (Fig 1C).

The telomeric probe hybridized all the telomeres found in the chromosomes (Fig 1D). Interstitial telomeric sequences (ITSs) were detected in the pericentromeric regions of the short arms of the homologs of pairs 1, 2, 7, and 10, and on the long arms of pairs 4 and 5. The FISH assays using microsatellite repeats $(GA)_{15}$ detected conspicuous hybridization signals in the terminal regions of the homologs of pair 1, as well as in a centromeric position of pairs 1, 2 and 9 (Fig 1E). Hybridization signals of the $(GA)_{15}$ probe were also observed in the terminal regions of pairs 2, 7, 8, 9, 10, and 12, in the terminal regions of pairs 3, 4 and 5 (Fig 1E). A strong signal was also observed in the long arm of pair 11 (Fig 1E). Pair 6 did not show hybridization signals with $(GA)_{15}$ probe (Fig 1E). Chromosome 1 presented hybridization signals of the $(GATA)_8$ probe in the terminal region of the long arm and terminal region of all chromosome pairs (Fig 1F). The location of the $(GACA)_4$ repeat coincided with the heterochromatic block in the centromere of chromosome 1 and in the euchromatic portion of the terminal region of its long arm, coinciding with the $(GA)_{15}$ signals (Fig 1G). Additional marks of this microsatellite were found in the terminal regions of all pairs, except 11 and 12 (Fig 1G). No

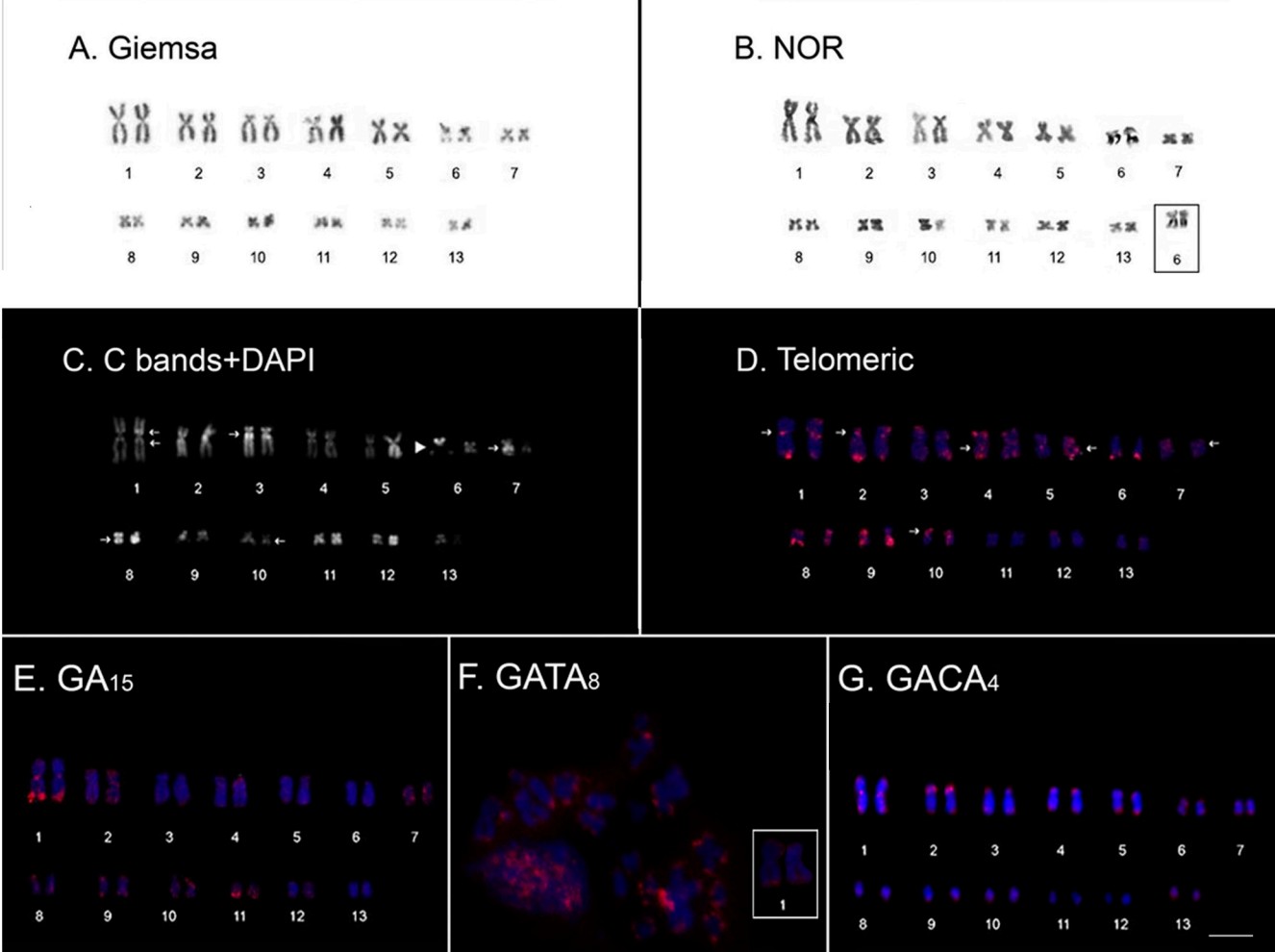

**Fig 1. Karyotype features and microsatellite motifs of *Cycloramphus bolitoglossus*.** Classical cytogenetic features are shown by Giemsa staining (**A**), location of the NORs, revealed by silver impregnation (**B**), and the heterochromatin revealed by C-banding + DAPI staining (**C**). Molecular cytogenetic mapping of the telomeric probe (**D**) and microsatellite motifs (GA)$_{15}$ (**E**), (GATA)$_8$ (**F**). and (GACA)$_4$ (**G**). Bar: 5μm.

hybridization signals were observed for the (CAG)$_{10}$, (CGC)$_{10}$, and (GAA)$_{10}$ microsatellite probes in this karyotype.

## Molecular analysis of the 5S rDNA of *Cycloramphus bolitoglossus*

The PCRs recovered two fragments of approximately 400 bps and 500 bps, with 12 recombinant colonies. All the cloned sequences were classified as the rDNA 5S type II, based on the BLAST searches of the NCBI nucleotide database, the similarity analysis with the 5S rRNA gene of other anurans (Table 1). Given the numerous differences in the *C. bolitoglossus* sequences, however, we classified them into two subtypes, IIa and IIb.

The 5S type IIa is the smaller of the two sequences (CB5STIIa.C1), which has 393 bps, including 120 bps that correspond to the coding sequence of the 5S rDNA gene and an NTS region containing 273 bps. The presumable transcribed region contains complete internal control regions, box A, intermediate element, and box C (Fig 2A). Eight sequences (CB5STIIb. C1-CB5STIIb.C4 and CB5STIIb.C6-C9) were identified as subtype "b", and contain 498 bps, with the first 120 bps corresponding to the complete coding sequence of the 5S rDNA gene

**Table 1. rDNA 5S genetic similarity.**

| Species and sequence | Similarity (%) | |
|---|---|---|
| | CB5STIIa | CB5STIIb |
| *C. bolitoglossus* Werner, 1897—CB5STIIa | - | 91.10 |
| *C. bolitoglossus* Werner, 1897—CB5STIIb | 91.10 | - |
| *Engystomops freibergi* Donoso-Barros 1969—5S II | 79.78 | 90.39 |
| *Engystomops petersi* Jiménes de la Espada 1872—5S II | 78.06 | 89.27 |
| *Physalaemus cuvieri* Fitzinger 1826—5S II | 77.03 | 88.07 |
| *Amolops mantzorum* David, 1872—5S II | 72.87 | 84.68 |
| *Gatrotheca riobambae* Fowler, 1913 | 72.87 | 84.68 |
| *Pseudis tocantins* Caramaschi and Cruz, 1998—5S II | 72.58 | 84.94 |
| *Anaxyrus americanus* Holbrook, 1836 | 71.10 | 83.11 |
| *Xenopus tropicalis* Gray, 1864 | 70.24 | 82.70 |
| *Pelophylax lessonae* Camerano, 1882 | 68.73 | 81.23 |
| *Pelophylax ridibundus* Pallas, 1771 | 68.73 | 81.23 |
| *Engystomops freibergi* Donoso-Barros, 1969—5S I | 68.65 | 81.19 |
| *Xenopus tropicalis* Gray, 1864—oocyte | 68.58 | 81.17 |
| *Amolops mantzorum* David, 1872—5S I | 68.29 | 80.86 |
| *Lithobates catesbeianus* Shaw, 1802 | 67.01 | 79.69 |
| *Engystomops petersi* 5S I | 66.78 | 79.57 |
| *Lithobates pipiens* Schreber, 1782 | 66.08 | 79.20 |
| *Xenopus borealis* Parker, 1936 | 64.93 | 77.29 |
| *Physalaemus cuvieri* Fitzinger 1826—5S I | 64.57 | 77.76 |
| *Xenopus laevis* Daudin, 1802 | 63.60 | 76.92 |
| *Xenopus laevis* Daudin, 1802—oocyte | 60.76 | 74.38 |
| *Pseudis tocantins* Jiménes de la Espada 1872—5S I | 39.45 | 55.84 |

Genetic similarity (%) between the sequences of rDNA 5S type I and II of *C. bolitoglossus* and other anuran species obtained from the GenBank database.

(including all internal regulatory elements) and an NTS region containing 378 bps (Fig 2B). We also recovered two cloned sequences. We designated CB5SPG (Fig 2C) and subsequently identified asthe PcP190 satDNA based on the BLAST searches of the NCBI sequence database and in our alignment with 5S rDNA and PcP190 SatDNA sequences from *C. bolitoglossus* (Fig 2D).

The nucleotide sequences of the presumable transcribed region (120 bps) of the 5S rDNA subtypes IIa and IIb diverge only slightly, with a mean similarity of 91.10% between them (Table 1). When sequences of this region, isolated from *C. bolitoglossus*, were compared with those of other anurans (Table 1), CB5STIIa presented the lowest similarities, ranging from 39.45% (with the *Pseudis tocantins* 5S I rRNA gene) to just under 80% in the case of *Engystomops freibergi* 5S rRNA gene type II. In contrast, the CB5STIIb sequence was just over 90% similar to the latter sequence (Table 1).

The Maximum Likelihood analysis grouped all the 5S rRNA gene sequences of *C. bolitoglossus* within a group with all the type II 5S rDNAs of *P. cuvieri*, *Engystomops*, and *Pseudis tocantins*. However, the CB5STIIa sequence was more divergent (Fig 3).

The chromosomal mapping of 5S rDNA subtype IIa (using the CB5STIIa.C1 clone) and subtype IIb (using the CB5STIIb.C1 clone) found distinct hybridization signals in two chromosomes pairs of male individuals of *C. bolitoglossus* (Fig 4A). The CB5STIIa.C1 probe detected a signal in the centromeric region of the homologs of pairs 1 and 4 (Fig 4B). In

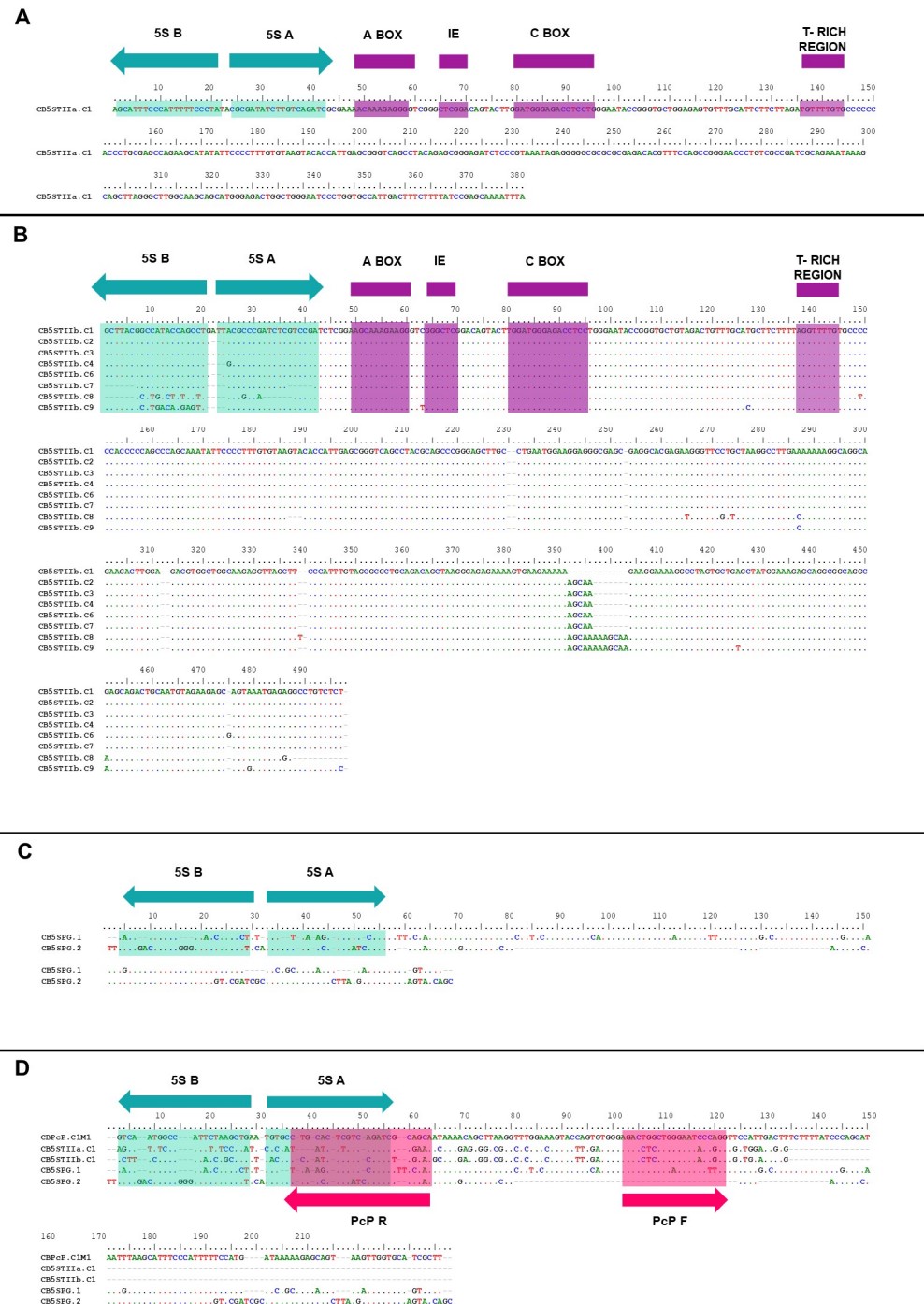

**Fig 2. 5S rDNA alignments.** Sequence alignments of the 5S rDNA clones obtained from *C. bolitoglossus* CB5STII.a (**A**), CB5STII.b (**B**), and CB5SPG (**C**). In **D**, CB5SPG is aligned with CB5STII.a, CB5STII.b, and the PcP190 satDNA. The primer annealing sites of 5S-A are highlighted as blue arrows whereas the PcP190 satDNA primer sites are highlighted as green arrows. The internal control regions of the 5S rDNA gene are shaded in purple in **A** and **B**.

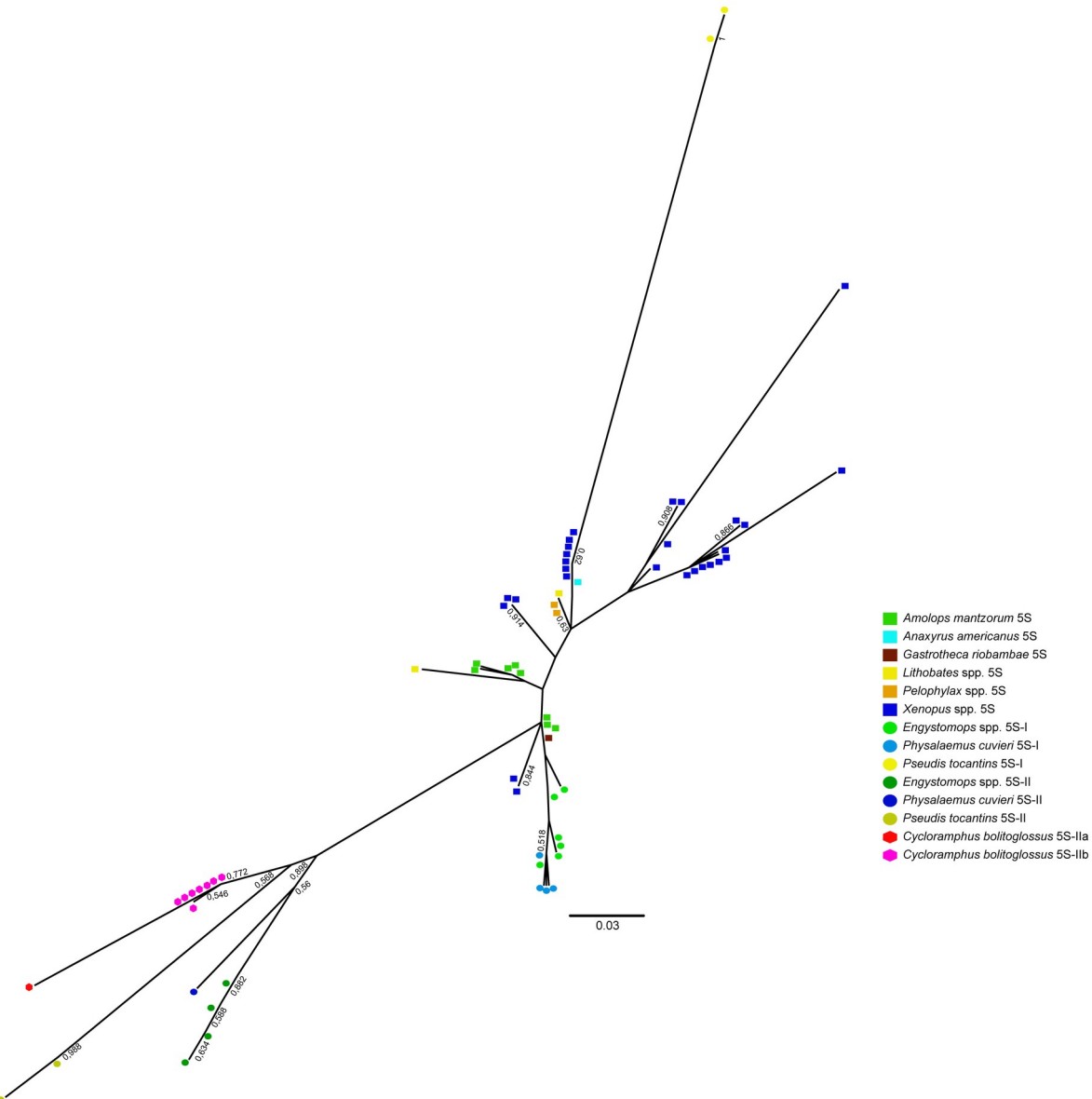

**Fig 3. 5S rDNA Maximum Likelihood analysis.** Arrangement derived from the Maximum Likelihood analysis of the 5S rRNA gene sequences of *Cycloramphus bolitoglossus* and other anurans obtained from GenBank.

contrast the CB5STIIb probe detected clusters of signals in the terminal region of the long arms of the homologs of pair 1 (Fig 4A).

## PcP190 satDNA identification reveals chromosome spreading in the *C. bolitoglossus* karyotype

The PCRs generated distinct bands ranging in length from 200 bps to 700 bps. The cloning produced nine recombinant clones, with four of these nine cloned sequences being composed of three monomers (607 bps) with juxtaposed sequences, while one clone presented one complete and one partial monomer (313 bps), and the other four recovered the partial PcP190 monomer (198 bps). All the monomers isolated in the same cloned sequence share an even

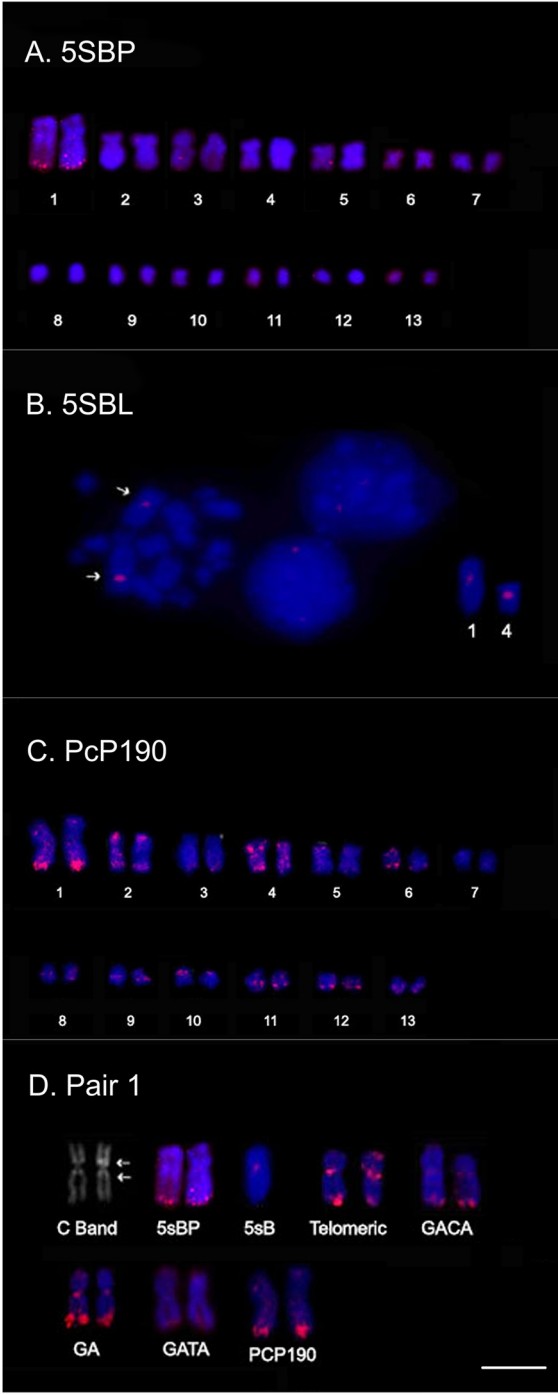

**Fig 4. Chromosomal mapping.** A. Location of CB5STII.b in the long arms of pair 1. B. Hybridization signal of CB5STIIa. C. Mapping of PcP 190 SatDNA. D. Comparative hybridization signals in chromosome pair 1 of *Cycloramphus bolitoglossus*. Bar: 5μm.

higher mean level of similarity (95.54%). We recognized conserved (CR) and hypervariable (HR) regions, based on the classification of Gatto et al. [25] (Fig 5).

Finally, we analyzed the similarity of the conserved region of the 5S rDNA clones and the CRs of the PcP clones of *C. bolitoglossus*. The cloned CB5STIIb sequences were almost 60%

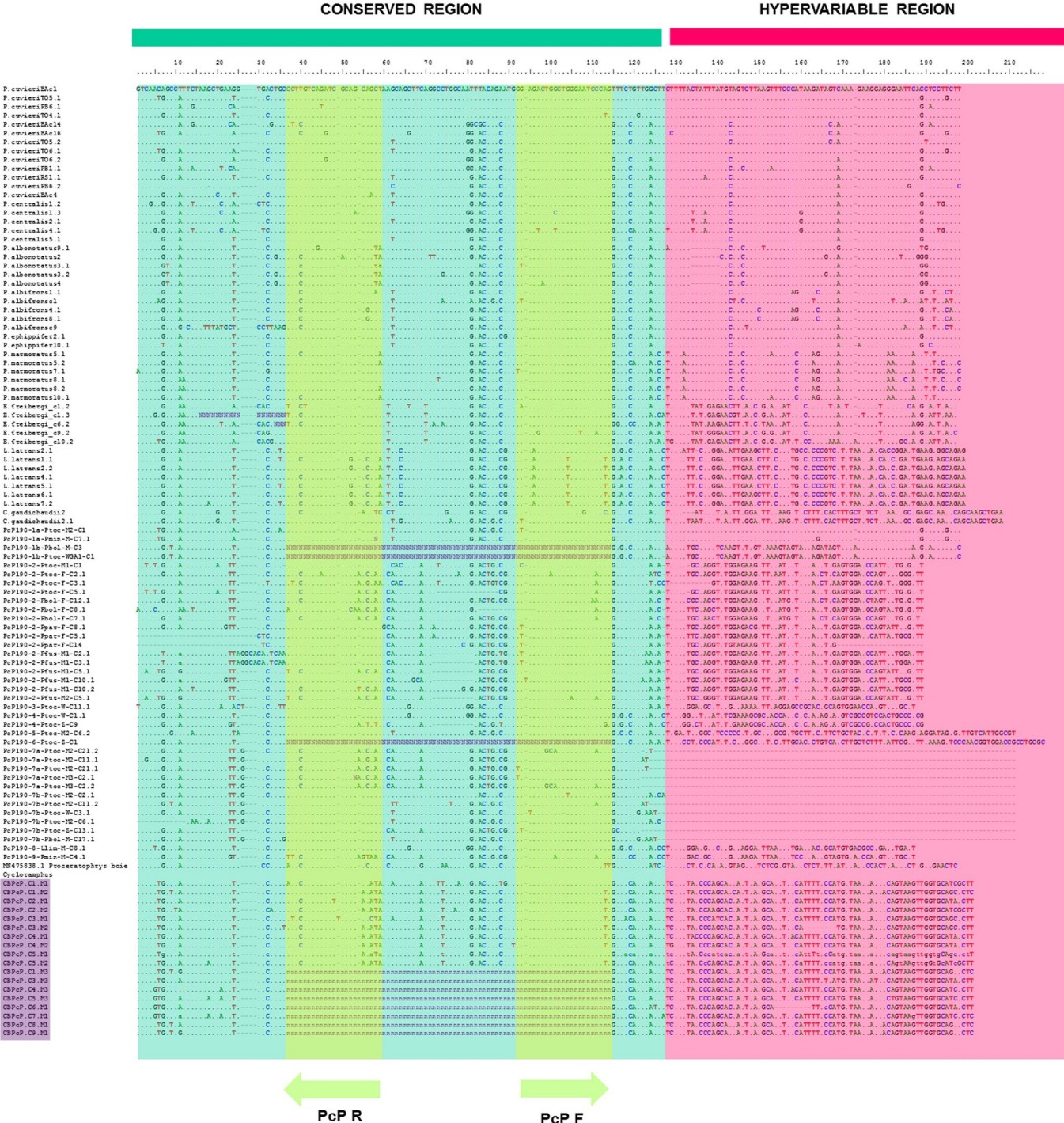

**Fig 5. PcP190 satDNA alignment.** Sequence alignment of the PcP190 satDNA clones obtained from the *C. bolitoglossus* genome (purple box) with the sequences from other anurans. Primer annealing sites are highlighted (green), as are conserved (blue) and hypervariable (pink) regions. The CR of the PcP190 of *C. bolitoglossus* was highly similar among the intragenomic monomers and in comparison with other anuran species (> 77%) deposited in GenBank (S4 Table). As expected, the HR presented low nucleotide variation at the intragenomic level (94.86%) (S4 Table), but high levels of variation in compared with the HRs of the other anurans included in the matrix.

similar to the CR of the PcP190 satDNA (S4 Table), although CB5STIIa was less similar, at 42.37%. The PcP190 satDNA probe produced highly scattered signals, with hybridization sites identified in the centromeric and telomeric regions of pairs 1, 2, and 4, and in the telomeric region of pairs 7, 8, 9, 10, 11, 12, and 13 (Fig 4C). In Fig 4D, we emphasized all hybridization marks found in pair 1.

## Discussion

### Insights into the evolution of the *Cycloramphus* species chromosomes

The karyotype of *C. bolitoglossus*, described here, presents a highly conserved diploid number of 2n = 26 chromosomes, as reported previously in other cycloramphids [5–8, 53]. Despite this conserved diploid number, interspecific variation in the fundamental number (FN = 50 or 52) indicates that intrachromosomal rearrangements may have played a central role in this genus chromosomal diversification.

The most comprehensive phylogenetic reconstruction of the genus *Cycloramphus* [9] found evidence of independent, recurrent events of change in the FN (Fig 6). If this hypothesis is correct, several pericentromeric inversions should be found in the different karyotypes identified in the phylogenetic tree of this genus. Chromosome pair 1 is a potentially good example of this process, given the existence of metacentric and subtelocentric morphology among different karyotypes (Fig 6).

When we adjusted the chromosome morphology of pair 1 to the phylogenetic inferences of de Sá et al. [9], we found evidence that this metacentric pair is the plesiomorphic condition in *Cycloramphus*. This conclusion is supported by the fact that pair 1 is metacentric in *Thoropa taophora* (the outgroup) [*T. miliaris* in 53] and *Zachaenus carvalhoi* [54, 55], and that this morphology is retained in *Cycloramphus brasiliensis*, *C. bolitoglossus*, *C. carvalhoi*, *C. eleutherodactylus*, *C. acangatan*, *C. boraceiensis*, *C. dubius*, *C. lutzorum*, and *C. rhyakonastes* [7, 54]. Given this evidence, the most parsimonious hypothesis to account for these morphological changes in the homologs of pair 1 would involve centromeric reposition events that modified the plesiomorphic (metacentric) condition, which would have occurred in the lineage that gave rise to *C. fuliginosus* [7].

Heterochromatic blocks are located in the pericentromeric regions of all the *C. bolitoglossus* chromosomes. However, *C. bolitoglossus* is different from the other congeners that have been karyotyped due to interstitial C-positive bands in the long arms of chromosomes 8 and 10. By contrast, *C. bolitoglossus* and *C. brasiliensis* do share a large amount of pericentromeric heterochromatin in the long arm of pair 3 [7], which indicate homology of pair 3 between these two species. Unfortunately, no C-banding data are available for *C. fuliginosus*, *C. migueli*, and *C. bandeirensis*. The collection of these data for these species would help resolve the homology hypothesis in pair 3 of these species.

Changes in the morphology of pairs 6 and 13 also appear to have involved centromeric reposition, resulting in variation in the metacentric or subtelocentric/telocentric morphology among the different species [7]. However, except *C. acangatan* and *C. fuliginosus*, all the *Cycloramphus* species that have been karyotyped to date [7, 53], including *C. bolitoglossus*, have rDNA 45S sites in pair 6, detected by Ag-NOR method. The rDNAs are considered a hot-spot of rearrangement [56], which may account for the changes observed in this chromosome pair among this genus species.

The chromosomal mapping of the repetitive DNA in the *C. bolitoglossus* karyotype permitted the assessment of these chromosomal changes from a new perspective. Our mapping data reveal the presence of rDNA 5S subtype IIa and satDNA PcP190 in chromosome pair 1 in addition to the enrichment of the microsatellite motifs [(GA), (GACA) and (GATA)] and the

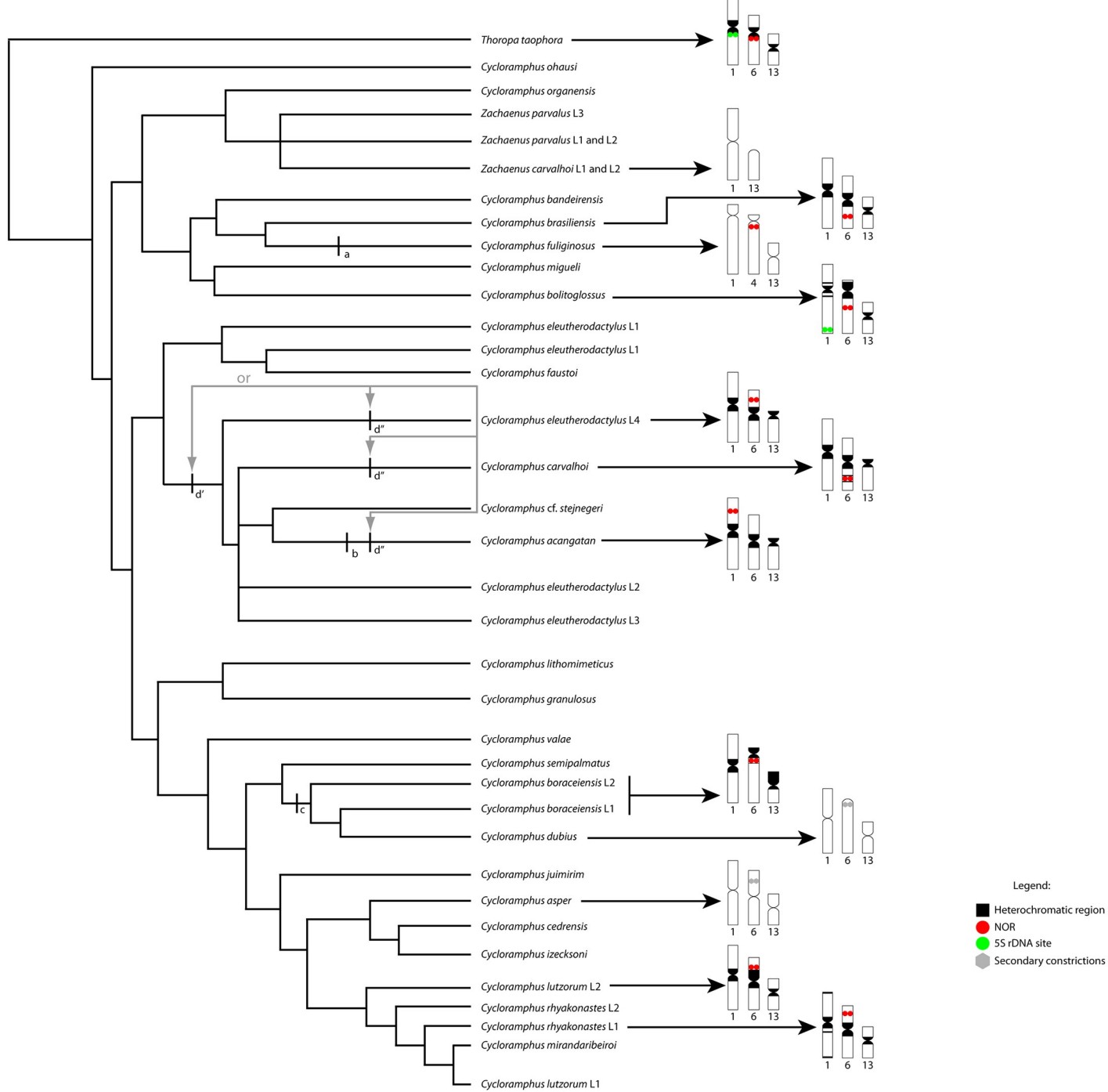

**Fig 6. Phylogenetic reconstruction of the genus *Cycloramphus*.** The hypothesis of the evolutionary chromosomal changes in the *Cycloramphus* and *Zachaenus* species, as inferred from the available cytogenetic data [5–8, 53] in the phylogeny of the genus presented by de Sá et al. [9]. Events of character transformation are indicated: (a) centromere repositioning in chromosome pair 1 of *C. fuliginosus*; (b) repositioning of the NOR from pair 6 to pair 1 in *C. acangatan*; (c) pericentromeric inversion in pair 6 of the common ancestor of the *C. boraceiensis* linages and *C. dubius* and (d' or d") a centromere repositioning in the chromosome 13 in the common ancestor of *C. eleutherodactylus* lineages L2, L3 and L4, *C. carvalhoi*, *C. stejnegeri* and *C. acangatan*, or independent repositioning in the lineages that gave rise to *C. eleutherodactylus* L4, *C. carvalhoi* and *C. acangatan*. The d' and d" events are equally parsimonious hypothesis (indicated by grey arrows) given the available cytogenetic data for the genus *Cycloramphus*. Red circles indicate the NORs positions, Gray circles indicate the presence of secondary constrictions, and black regions indicate heterochromatin areas of the chromosomes.

presence of interstitial telomeric sequences (ITSs) in this pair. Microsatellites are typically associated with regions that have high recombination rates, and it is thought that the microsatellites $(GA)_{15}$ and $(GACA)_4$ are related to the stabilization of the ribosomal DNA sequences [29]. Given this, the comparative cytogenetics of *Cycloramphus* revealed an unexpected model for the investigating of the role of the repetitive DNAs in the intrachromosomal rearrangements observed in the present study.

## Recurrent association between satDNA PcP190 and rDNA 5S in the Anura: Additional evidence

We recorded satDNA PcP190 in the family Cycloramphidae for the first time, expanding the known occurrence of this family of satDNA in the anuran genome. The PcP190 sequences of *C. bolitoglossus* were most similar to the first 120 bps of the other anuran PcP190 sequences, consistent with the conserved domain (CR) already reported for this satDNA family [23–25]. A hypervariable region was also recognized within the PcP190 monomer that did not match any other PcP190 deposited in GenBank. Given this, we suggest that the PcP190 of *C. bolitoglossus* represents a new type of sequence of this satDNA family, denominated as PcP190-Cbol-1.

When we compared the satDNA PcP190 among the anuran species in which this sequence has already been reported, several similar patterns can be observed in this repetitive DNA, including (i) high intra- and intergenomic levels of similarity, (ii) the chromosomal distribution in the FISH assays, and (iii) the recurrence of associations with the 5S rDNA chromosomal clusters. Firstly, despite the dispersal of the PcP190-Cbol-1 satDNA copies among the *C. bolitoglossus* chromosomes, we recovered an unexpectedly high degree of nucleotide similarities among the cloned sequences in both the CR and the HR. The high level of homogenization observed in the *C. bolitoglossus* genome may reflect the influence of the concerted evolution model for this repetitive DNA or a single recent event of expansion in this genome [57, 58]. It may also be possible that the PCR isolated only highly homogeneous sequences from the heterochromatic regions because they are more abundant. It is interesting to note that, in *C. bolitoglossus*, the FISH mapped the PcP190-Cbol-1 satDNA in the euchromatic areas. This was unexpected because the satDNA elements distributed in euchromatic domains tend to be more widely dispersed then those located in heterochromatic regions [59].

The chromosomal mapping of satDNA PcP190 in anuran karyotypes typically reveals a long array of juxtaposed monomeric units in the genome, located primarily in the heterochromatic regions [22, 24], as detected in the terminal and/or (peri)centromeric regions of the *C. bolitoglossus* chromosomes with a probe for PcP190-Cbol-1. This repetitive DNA has also been mapped in the (peri)centromeric heterochromatin of several *Physalaemus* species [22, 23, 60] and a heterochromatic block of the W chromosome of *Pseudis tocantins* [24]. Thus, the occurrence of the PcP190 in heterochromatin regions is common and broadly expected.

However, our FISH assays also revealed PcP190-Cbol-1 satDNA in the euchromatic regions of the *C. bolitoglossus* chromosomes, a situation that was not found in any other species investigated so far that possess PcP190 satDNA [23–25]. A similar pattern has already been detected to the satDNA families in the karyotypes of *Drosophila melanogaster* Meigen 1830 [61] and the red flour beetle, *Tribolium castaneum* Herbst 1797 [62], including the transcriptional regulation of genes associated with the euchromatic arrays of the TCAST1 satDNA, which modulate gene expression under stress in *T. castaneum* [63]. The functional significance of the PcP190 satDNA has never been verified in the anuran genomes. However, the highly conserved sequence found in numerous distantly-related species of the Hyloidea superfamily supports the hypothesis that the PcP190 satDNA could have a functional role in the hyloid genome [23–

25]. Given this, the detection of PcP190 satDNA in euchromatic regions of the *C. bolitoglossus* chromosomes indicates that this species may represent a potentially valuable model species for verifying the functional role of this satDNA in the future.

As observed in the local association of the 5S rDNA subtype IIa and satDNA PcP190 in the terminal region of chromosome pair 1, the apparent interplay between repetitive classes does not appear to be random. The evolutionary derivation of the satDNA PcP190 found in anurans from the rDNA 5S is supported by the high degree of similarity in the conserved 120 bps of the rDNA gene and the CR of the satDNA. Vittorazzi et al. [22] suggested that the PcP190 of *Physalaemus cuvieri* originated from the duplication of genes following the divergence and dispersion to new chromosome sites. Similar associations have also been reported in *P. ephippifer* [23, 64], *Engystomops petersi*, *E. freibergi*, and *E.* cf. *magnus* [10]. We also detected a similar pattern in the present study, with the major cluster of the PcP190 satDNA coinciding with 5S rDNA subtype IIa in *C. bolitoglossus*, in addition to signals dispersed in subtelomeric or (peri) centromeric regions of other chromosomes, corroborating the hypothesis of Vittorazzi et al. [22].

Additionally, interesting to note that the 5S rDNA was mapped in chromosome 1 of *C. bolitoglossus*. In the genus *Thoropa* (which is closely related to *Cycloramphus*) the large metacentric chromosome 1 also showed to harbor the 5S rDNA sites in most of the species [53]. Based on this, we may infer that chromosome 1 of *C. bolitoglossus* might be homologous to chromosome 1 from *Thoropa* species.

The 5S rDNA analyses of *C. bolitoglossus* recovered two different sizes of 5S rDNA sequences as found in some other species [17, 65, 66]. The difference between the two products' molecular weights has been attributed in particular to the differences in the size and composition of the NTS, which is a diversified sequence under low selective pressure, and thus more susceptible to the accumulation of mutations [67]. Despite its potential as a cytogenetic marker, 5S rDNA has been sequenced and mapped in only a few anuran species [65, 68, 69].

The Maximum Likelihood analysis grouped all the *C. bolitoglossus* 5S rRNA gene sequences in the same cluster, distinct from the 5S rDNA type II of other anurans. This indicates a high degree of homogenization of the 5S rRNA gene in *C. bolitoglossus*. However, the "a" subtype of the *C. bolitoglossus* 5S rDNA was more divergent than the "b" subtype in both the maximum likelihood and similarity analyses. This indicates conclusively that the two subtypes may represents distinct evolutionary pathways or even reflect a process of pseudogenization. It has been suggested that multigene families evolve through a birth-and-death model of evolution, which might explain why duplications events often lead to the duplication of copies of a given sequence, which would not evolve in concert with other parental copies, but might become a new gene or even degenerate into a pseudogene [18].

Pseudogenes of the 5S rDNA have already been identified in some ray-finned fish, Actinopterygii [70, 71], human and rat genomes [72, 73]. Some satDNAs sequences are thought to have been derived from 5S rDNA by an ancient pseudogenization process, including the 5S-*Hind*III in the fish *Hoplias malabaricus* Bloch 1794 [74], PLsatB in the weed *Plantago lagopus* Linneaus 1753 [75], and the PcP190 in hyloid anurans [23–26]. In the present study, we used the 5S-A and 5S-B primers of Pendás et al. [43], which are universal primers used to isolate 5S rDNA in vertebrate genomes [21, 24, 67, 70, 71]. However, two cloned sequences (CB5SPG.1 and CB5SPG.2) were highly similar to the PcP190 satDNA sequences in the BLAST searches, indicating that they are, in fact, PcP190 satDNA. These results reinforce the similarities in the 5S rDNA and PcP190 satDNA sequence, given that the primers of Pendas et al. [43] may anneal in the regions of 24–44 bps (5S-A) and 1–21 bps (5S-B) in the PcP190 satDNA copies. Given this, future studies of 5S rDNA in hyloid frogs that use the universal primers of Pendas et al. [43] should pay careful attention after the cloning or sequencing procedures to avoid

cross-amplification with PcP190 satDNA. Overall, then, the evidence on pseudogenization and the emergence of satDNA from rDNA 5S clusters, reported here, further reinforce the importance of the birth-and-death evolutionary model to explain the rDNA 5S patterns found in anuran genomes.

## Conclusions

The use of novel chromosomal markers provided new perspectives for the investigation of chromosomal evolution in the genus *Cycloramphus* and suggests, in particular, that repetitive DNAs may have played an important role in karyological diversification without chromosome number changes. Our data shed new light onto the evolutionary interplay between satDNA PcP190 and rDNA 5S in the anuran genome and raise further questions on its evolutionary significance.

## Supporting information

**S1 Table. 5S rDNA accession numbers.**
(DOCX)

**S2 Table. PcP190 satDNA accession numbers.**
(DOCX)

**S3 Table. Chromosome measurements.**
(DOCX)

**S4 Table. PcP190 SatDNA genetic similarity.**
(DOCX)

## Acknowledgments

We thank the Multi-User Confocal Microscopy Center of the Federal University of Paraná for the capture of the images included in this study. We thank the Fundação Grupo O Boticário de Proteção à Natureza and Mater Natura—Instituto de Estudos Ambientais for logistical support in collections.

## Author Contributions

**Conceptualization:** Daniel Pacheco Bruschi.

**Data curation:** Gislayne de Paula Bueno, Kaleb Pretto Gatto.

**Formal analysis:** Gislayne de Paula Bueno, Kaleb Pretto Gatto, Camilla Borges Gazolla.

**Investigation:** Gislayne de Paula Bueno, Kaleb Pretto Gatto, Peterson T. Leivas, Michelle M. Struett, Maurício Moura.

**Methodology:** Gislayne de Paula Bueno, Camilla Borges Gazolla.

**Project administration:** Daniel Pacheco Bruschi.

**Supervision:** Daniel Pacheco Bruschi.

**Validation:** Gislayne de Paula Bueno.

**Writing – original draft:** Gislayne de Paula Bueno, Kaleb Pretto Gatto, Peterson T. Leivas, Michelle M. Struett, Maurício Moura, Daniel Pacheco Bruschi.

**Writing – review & editing:** Gislayne de Paula Bueno, Kaleb Pretto Gatto, Camilla Borges Gazolla, Maurício Moura, Daniel Pacheco Bruschi.

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
