## [Decision Letter · Decision Letter 0]

14 Oct 2020

PONE-D-20-23522

Chromosomal mapping of the repetitive DNAs of the karyotype of Cycloramphus bolitoglossus, an endemic frog from the Brazilian Atlantic Forest biome (Anura, Cycloramphidae)

PLOS ONE

Dear Dr. Bruschi,

Thank you for submitting your manuscript to PLOS ONE. After careful consideration, we feel that it has merit but does not fully meet PLOS ONE’s publication criteria as it currently stands. Therefore, we invite you to submit a revised version of the manuscript that addresses the points raised during the review process.

As can you see below, your paper was revised by three reviewers. One of them reject and the other two supposes that your manuscript needs a revision. After a careful reading of the manuscript, my decision is a major revision of your paper. I suggest you to do a thorough review of English, and answer all the reviewers’ questions in detail (Note that reviewer 2 has included an attachment with their comments). Please review the PCR conditions, especially the concentrations of the reactions. I recommend that phylogenetic reconstructions to be carried out in a more robust program, for example RAxML. Some figures are not mentioned in the text and others are misreferred, you should check all of them carefully. Please, I would like you to know that the final acceptance of your manuscript will depend on the quality of the review of your manuscript and the responses to the reviewers' comments. Please let me know if you have any questions.

We look forward to receiving your revised manuscript.

Kind regards,

Maykon Passos Cristiano, D.Sc.

Academic Editor

PLOS ONE

Journal Requirements:

2. Please amend either the title on the online submission form (via Edit Submission) or the title in the manuscript so that they are identical.

We thank the Federal University of Paraná (UFPR), the Brazilian National Council for

475 Scientific and Technological Development (CNPq) for PIBIC fellowships to GPB and VB, and

476 the Coordination for Higher Education Personnel Training (CAPES/PROAP – Finance Code

477 001) for the scholarships provided to CBG and MMS. KPG is grateful to the São Paulo State

478 Research Foundation (FAPESP) for a postdoctoral fellowship (grant 2018/17992-0). We thank

479 the Multi-User Confocal Microscopy Center of the Federal University of Paraná for the capture

480 of the images included in this study. Fieldwork was partially funded by Fundação Grupo O

481 Boticário de Proteção à Natureza and conducted by Mater Natura - Instituto de Estudos

482 Ambientais).

i) We note that you have provided funding information that is not currently declared in your Funding Statement. However, funding information should not appear in the Acknowledgments section or other areas of your manuscript. We will only publish funding information present in the Funding Statement section of the online submission form.

ii) Please remove any funding-related text from the manuscript and let us know how you would like to update your Funding Statement. Currently, your Funding Statement reads as follows:

" The author(s) received no specific funding for this work.".

5. Please upload a new copy of Figure 5 as the detail is not clear. Please follow the link for more information: https://blogs.plos.org/plos/2019/06/looking-good-tips-for-creating-your-plos-figures-graphics/" https://blogs.plos.org/plos/2019/06/looking-good-tips-for-creating-your-plos-figures-graphics

<h1>** **</h1>

Reviewers' comments:

Reviewer's Responses to Questions

**Comments to the Author**

1. Is the manuscript technically sound, and do the data support the conclusions?

Reviewer #1: No

Reviewer #2: Yes

Reviewer #3: Yes

2. Has the statistical analysis been performed appropriately and rigorously? 

Reviewer #1: N/A

Reviewer #2: N/A

Reviewer #3: Yes

3. Have the authors made all data underlying the findings in their manuscript fully available?

Reviewer #1: Yes

Reviewer #2: Yes

Reviewer #3: Yes

4. Is the manuscript presented in an intelligible fashion and written in standard English?

Reviewer #1: No

Reviewer #2: Yes

Reviewer #3: Yes

5. Review Comments to the Author

Reviewer #1: The authors performed some molecular cytogenetic analysis in the frog Cycloramphus bolitoglossus. The main goal of the work “to understand the chromosome shifts that have occurred during the evolutionary history of this group” is unfortunately not fully achieved. The authors analyze just one species and made several phylogenetic inferences about possible evolutionary pathways inside the genus, but as all closely related species lack similar cytogenetic analysis (as stated by the authors lines 82-84), the conclusions are too speculative. “The use of novel chromosomal markers provided new perspectives for the investigation of chromosomal evolution in the genus Cycloramphus”: In fact, they CAN add a new chapter into this story IF also applied to more species. Why haven´t the authors perform such analysis in more species from this genus? Besides, Title is very descriptive, since it must contain the real findings of this papers. In the same way abstract is also very descriptive without a clear central line of objective/conclusions. Some figures are also strange/incomplete and must be arranged: eg. Fig1Dand Fig4A the pair are not properly aligned; Fig 1C, why there is a bigger chromosomal pair named as “6” in the end of the karyotype? Fig1F is probably both incomplete and of terrible quality and must be replaced. And to finish, English writing is also considerably problematic and needs extensive revision before any new attempt of submission.

Reviewer #2: Dear editor,

This study presents the karyotype description and classical and molecular cytogenetic patterns Cycloramphus bolitoglossus, endemic species of the Brazilian Atlantic Forest. This is a useful and informative paper, discussing important information about the group's chromosomal Evolution.

Reviewer #3: This work includes a study on the cytogenetics of Cycloramphus bolitoglossus. The study describes for the first time the karyotype of the species, with conventional and several molecular approaches, isolating and mapping several classes of repetitive DNA elements by FISH. Moreover, this study provides more empirical information on the possible origin of the satellite PcP190, by pseudogenization of 5S rDNA. This paper is original and will be a valuable contribution to the genus and also for Cycloramphidae.

I found some issues and gave additional suggestions above to the authors with the intention of improving the manuscript. In this sense, I consider that they deserve to be attended before the manuscript is fully appropriate for publication.

Abstract.

Line 51- ...’the pattern of DNA repetitive classes’… In my opinion, it should be added the word ‘some’ before DNA, as in this study are analyzed only two repetitive DNA classes plus seven microsatellites.

Line 55- Please, provide the FN of C. bolitoglossus in the abstract.

Lines 64-67- The authors should add a brief description here of PcP190 occurrence in other groups of Anura, or at least state that this sequence generally shows a hypervariable region and a more conserved one.

Line 67- ”our data showed/have shown evidence of”? Please check this sentence.

Keywords.

The word repetitive DNA is on the title and may be replaced, although keywords are not included in the final version of the manuscript.

Introduction.

Line 82- I think that it is more appropriate to state the number of studied species instead of ‘Few’.

Line 84- The diploid number (2n=26) is observed in all species. The word “almost” should be removed or the sentence modified, as some species are showing non bi-armed chromosomes.

As a suggestion: “The diploid number (2n=26) is conserved/observed in all the species, with karyotypes that are generally composed of metacentric and submetracentic pairs”.

Line 85- the word submetacentric is misspelled.

Lines 87-88- To date, there were studied ten species of Cycloramphus (Noletto et al., 2011). Three of them showed FN=50 while the others a FN=52. Other species with FN=52 should be also included in this statement as there were named only two of them.

Lines 95-101. Why including 45S in this paragraph? Although being rDNA this sequence was not studied here and should be removed.

Material and Methods.

Line 139- Specimens were deliberately collected, independently of their low abundance. The word “accidentally” should be removed.

Line 154- Howell is misspelled

Line 155- Sumner is misspelled

Line 166- The PcP190 sequence was also amplified with this protocol; it should be included here or describe the PCR protocol used for PcP190. Moreover, if possible, provide the annealing temperatures used for both sequences.

Line 167- “110x PCR Buffer”. Please correct this, and the word buffer should be without a capital letter.

Lines 173-174- Please provide the manufacturers of these kits.

Lines 177 and 178- The words forward and reverse, including colons, should be without italics.

Lines 188-189- I think that comparisons between these four groups of sequences are not exclusively restricted to C. bolitoglossus, are they? If not, please state if in a different manner.

Lines 202, 209- ”(Roche)”. In order to follow the same criterion, please provide the country information as before.

Line 2013- same comment for Sigma-Aldrich.

Line 220- “…(0.4μg/Ml)”, please correct “Ml”.

Results.

Lines 229-231. At this point, a question arises. How did you measure chromosomes in order to determine their chromosome morphology and what criterion did you adopt? I recommend measuring them using one of the non-commercial software Drawid or Micromeasure, or other, and use the nomenclature proposed by Green and Session (1991) for anuran chromosomes.

Without measurements, chromosome morphology becomes very subjective and can affect phylogenetic interpretations.

For example, in my opinion, chromosomes of pair 1 are “more metacentric” than those chromosomes of pair 5. However, you considered them as submetacentric and metacentric, respectively.

Moreover, in the Discussion section (Lines 329-331), it is discussed about a metacentric morphology for pair 1 would be plesiomorphic for Cycloramphus.

Although results and interpretations would possibly remain unchanged, this is important for using the same criteria among cytogeneticists, and sometimes also provide interesting subtle differences when comparing species karyotypes.

Line 233- Figure 1C is referenced before Figure 1B (line 237). This must be corrected.

Lines 233-238- The authors applied DAPI after C-bands in order to enhance resolution. In my experience, this works, although this method allows detecting bands that sometimes are undetectable by the conventional C-band protocol, for example, the conspicuous heterochromatic bands observed in the pericentromeric region of pair 1.

As a suggestion, to avoid future inconsistencies in the literature or errors in the interpretation of chromosomal characters in other studied species Cycloramphus, authors should emphasize in this paragraph that these results correspond to C-bands+DAPI (i.e. DAPI positive marks after C-bands). Moreover, the first and only mention in this paragraph of the fluorochrome DAPI is between Lines 237-238, stating that “Negative DAPI staining was also observed in pair 6…”.

Lines 245-246- “Chromosome 1 also presented hybridization signals of the (GATA) 8 probe in the terminal region of the long arm (Figure 1D).”

However, (GATA)8 FISH is not shown, and the Figure 1D is the FISH with the telomeric probe. I am right?

Lines 248-249- “The telomeric probe hybridized all the telomeres found in the chromosomes (Figure 1D).”

Please check Figure 1 citing order. Regardless of whether Figure 1 has been cited as a whole, Figure 1D must be quoted before in the text or change it reference order. The same to Figure 1B.

Line 250- I can't visualize some of the ITSs bands described or distinguish them from other non-ITS marks. Is there any possibility to increase the quality of this figure or use insets of the described ITS bands? For instance, the pericentromeric ITS described for homologs of pair 10. Is it possible that the left chromosome of this pair is inverted?

Line 252- From which specimen/s did the 5S and PcP190 sequences come from? All specimens where molecularly studied? I can not find in the text, and I think is relevant or at least knowing the sex of the specimens from which the sequences come.

Table 1. Why not sorting this table in ascending / descending order of similarity (%)?

Lines 287-288- “The CB5STIIa.C1 probe detected a signal in the centromeric region of the homologs of pairs 1 and 4 (Figure 4B),…”

Again, when detecting variation, in my opinion, it is important to clarify in which sample or at least indicate in the legend of the figure in which sex the experiments were carried out. This has the goal that this information is explicitly established for future studies.

S3 Table Remarks. Should be PcP190 or PcP-Cbol instead of PcP.

Discussion

Lines 329-331- How was adjusted the morphology of chromosomes of pair 1 or what criterion did you follow? Again, as I recommended in the Results section, without measuring chromosomes, it is hard to speculate. Why not considering pair 1 of C. bolitoglossus as metacentric at first and avoid this speculation? In my opinion, the most conspicuous difference regarding pair 1 is that one observed in C. fuliginosus by Noleto et al., which seems to be an autapomorphic trait. Moreover, as you stated, the metacentric morphology is apparently homeologous among species and plesiomorphic.

Lines 335-338- I disagree in part with this statement. I cannot perceive a difference between chromosomes of pair 1 shown in this study for C. bolitoglossus than those present in C. acangatan and C. eleutherodactylus for example, which are ‘metacentric’.

Line 380- At this point I have a question regarding the nomenclature of Sat DNAs. First, in other species it is often used a nomenclature that denotes Species_MonomerLength (e.g., PcP190). Second, the sequence found in this species shows > 80% similarity with those found in other anurans, which possibly corresponds to a new variant of PcP190. Why naming this new variant as PcP-Cbol-1 instead of, for example, PcP190-Cbol-1?

Lines 339-402- Interesting. PcP190 usually has a centromeric position and, if I am not wrong, this is the first time that this sequence maps on a non-centromeric region of a ‘non-sex’ chromosome. Maybe, this should be highlighted.

Fig 1. B. Should say: C-Bands + DAPI

6. PLOS authors have the option to publish the peer review history of their article (what does this mean?). If published, this will include your full peer review and any attached files.

Reviewer #1: No

Reviewer #2: No

Reviewer #3: No

---

## [Author Response · Author response to Decision Letter 0]

6 Nov 2020

Our responses to the suggestions of the reviewers are presented in Respond to reviewrs cover. We would like to thank all the reviewers for their important contributions for the improvement of the manuscript. The corrections are highlighted in the text of the manuscript.

 Cordially,

Daniel Pacheco Bruschi

---

## [Decision Letter · Decision Letter 1]

27 Nov 2020

PONE-D-20-23522R1

Cytogenetic characterization and mapping of the repetitive DNAs in Cycloramphus bolitoglossus (Werner, 1897): more clues for the chromosome evolution in the genus Cycloramphus (Anura, Cycloramphidae)

PLOS ONE

Dear Dr. Bruschi,

Thank you for submitting your manuscript to PLOS ONE. After careful consideration, we feel that it has merit but does not fully meet PLOS ONE’s publication criteria as it currently stands. Therefore, we invite you to submit a revised version of the manuscript that addresses the points raised during the review process.

I read your manuscript again and recommend another round of minor revision. I found some small details that need to be improved before a decision.

We look forward to receiving your revised manuscript.

Kind regards,

Maykon Passos Cristiano, D.Sc.

Academic Editor

PLOS ONE

Additional Editor Comments (if provided):

My comments to improve the manuscript are as follows:

- Include authority and year in the taxa the first time it appears in the text.

- (Page 6, lines 132-133): “The chromosomes were C-banded, following Sumner [40], to define the heterochromatic pattern. This sentence is awkward, please improve it.

- (Page 15, line 337): What does it mean? Ssp is species or specimens?

- (Page 16, line 356): I did not find in the figure (d and d’).

- (Page 17, lines 383-385): “It is interesting to note, however, except C. acangatan and C. fuliginosus, all the Cycloramphus species that have been karyotyped to date…”. This sentence is awkward, please improve it.

- (Page 19, line 452): Why is “magnus” in quotation marks? I recommend that you check this information, and if necessary use “cf.” or “aff.” in the taxonomic context.

- (Page 21, line 480): Please remove “and even”.

-S3 Table: please check English in the legend of table S3

Please address minor comments by the reviewer #3.

Reviewers' comments:

Reviewer's Responses to Questions

**Comments to the Author**

1. If the authors have adequately addressed your comments raised in a previous round of review and you feel that this manuscript is now acceptable for publication, you may indicate that here to bypass the “Comments to the Author” section, enter your conflict of interest statement in the “Confidential to Editor” section, and submit your "Accept" recommendation.

Reviewer #1: All comments have been addressed

Reviewer #2: All comments have been addressed

Reviewer #3: All comments have been addressed

2. Is the manuscript technically sound, and do the data support the conclusions?

Reviewer #1: Yes

Reviewer #2: Yes

Reviewer #3: Yes

3. Has the statistical analysis been performed appropriately and rigorously? 

Reviewer #1: Yes

Reviewer #2: N/A

Reviewer #3: N/A

4. Have the authors made all data underlying the findings in their manuscript fully available?

Reviewer #1: Yes

Reviewer #2: Yes

Reviewer #3: Yes

5. Is the manuscript presented in an intelligible fashion and written in standard English?

Reviewer #1: Yes

Reviewer #2: Yes

Reviewer #3: Yes

6. Review Comments to the Author

Reviewer #1: The authors have nicely reviewed the MS and it can now be considered for publication. Although I still believe that the quality of some metaphases are very bad, I understood the sampling and conservation issues stated by the authors and this must be taken into consideration.

Reviewer #2: Dear editor,

This study presents the karyotype description and classical and molecular cytogenetic patterns Cycloramphus bolitoglossus, endemic species of the Brazilian Atlantic Forest. This a useful and informative paper discussing important information about the group's chromosomal Evolution. After the suggestions and corrections the manuscript became more informative. The authors made all the suggested corrections. The objectives have been answered and the article is ready for publication.

Reviewer #3: I consider that the authors attended all my suggestions, and this new version of the manuscript has improved. Therefore, I recommend accepting this manuscript.

However, during this new revision, I did detect some minor issues that must be corrected.

Lines 61-62: “C.boraceiensis, C.dubius”… add a space after “C.”.

Line 69: This line break space should be removed.

Lines 136-137: Drawid … [41] and Green and Session [42] references do not match the numbers in the Reference section. I think that this could be because Drawid’s reference is missing in the Reference section. Please, I strongly recommend rechecking all references in the text but also in the Reference section.

Line 236: “…4 and 5.The…”. Please add a space before “The”

Line 337: As a suggestion, replace “spp.” with “species”

Line 379: “…C. bandeirensisThe …”. Please correct this issue.

Line 392: “sequences (ITSs) in of this pair..”. Please correct this sentence.

Line 433: “C. bolitoglossus chromosomes,a”. Please correct this.

Line 463:”…67].The …”. Please correct this.

Line 466: “…[68]. . Despite its…”. Please correct this.

Line 626: This line break space should be removed

Line 677: Please replace reference number “584” with “58”.

7. PLOS authors have the option to publish the peer review history of their article (what does this mean?). If published, this will include your full peer review and any attached files.

Reviewer #1: No

Reviewer #2: No

Reviewer #3: No

---

## [Author Response · Author response to Decision Letter 1]

10 Dec 2020

Author response

We attended to all suggestions of the editor and reviewers. We thank all of you for the contributions for the improvement of the manuscript. The corrections are highlighted in the text of the manuscript with track changes.

Cordially,

Daniel Pacheco Bruschi

Editor Comments

 Include authority and year in the taxa the first time it appears in the text.

R: Suggestion accepted. We corrected in the new version of the manuscript.

Page 6, lines 132-133: “The chromosomes were C-banded, following Sumner [40], to define the heterochromatic pattern. This sentence is awkward, please improve it.

R: Sentence was reviewed.

Page 15, line 337: What does it mean? Ssp is species or specimens?

R: It means species. We corrected in the new version of the manuscript.

Page 16, line 356: I did not find in the figure (d and d’).

R: We corrected in the new version of the figure in the manuscript.

Page 17, lines 383-385: “It is interesting to note, however, except C. acangatan and C. fuliginosus, all the Cycloramphus species that have been karyotyped to date…”. This sentence is awkward, please improve it.

R: Sentence was reviewed. 

Page 19, line 452: Why is “magnus” in quotation marks? I recommend that you check this information, and if necessary use “cf.” or “aff.” in the taxonomic context.

 R: Suggestion accepted. We corrected in the new version of the manuscript.

Page 21, line 480: Please remove “and even”.

R: Suggestion accepted. We corrected in the new version of the manuscript.

S3 Table: please check English in the legend of table S3

 R: Sentence was reviewed.

Reviwer #3

Lines 61-62: “C.boraceiensis, C.dubius”… add a space after “C.”.

R: We corrected in the new version of the manuscript.

Line 69: This line break space should be removed.

R: We corrected in the new version of the manuscript.

Lines 136-137: Drawid … [41] and Green and Session [42] references do not match the numbers in the Reference section. I think that this could be because Drawid’s reference is missing in the Reference section. Please, I strongly recommend rechecking all references in the text but also in the Reference section.

R: We corrected this . We also checked all references in the new version of the manuscript.

Line 236: “…4 and 5.The…”. Please add a space before “The”

R: We corrected in the new version of the manuscript.

Line 337: As a suggestion, replace “spp.” with “species”

R: We corrected in the new version of the manuscript.

Line 379: “…C. bandeirensisThe …”. Please correct this issue.

R: We corrected in the new version of the manuscript.

Line 392: “sequences (ITSs) in of this pair..”. Please correct this sentence.

R: We corrected in the new version of the manuscript.

Line 433: “C. bolitoglossus chromosomes,a”. Please correct this.

R: We corrected in the new version of the manuscript.

Line 463:”…67].The …”. Please correct this.

R: We corrected in the new version of the manuscript.

Line 466: “…[68]. . Despite its…”. Please correct this.

R: We corrected in the new version of the manuscript.

Line 626: This line break space should be removed

R: We corrected in the new version of the manuscript.

Line 677: Please replace reference number “584” with “58”.

R: We corrected in the new version of the manuscript.

---

## [Editor Report · Decision Letter 2]

16 Dec 2020

PONE-D-20-23522R2

Cytogenetic characterization and mapping of the repetitive DNAs in Cycloramphus bolitoglossus (Werner, 1897): more clues for the chromosome evolution in the genus Cycloramphus (Anura, Cycloramphidae)

PLOS ONE

Dear Dr. Bruschi,

Thank you for submitting your manuscript to PLOS ONE. After careful consideration, we feel that it has merit but does not fully meet PLOS ONE’s publication criteria as it currently stands. Therefore, we invite you to submit a revised version of the manuscript that addresses the points raised during the review process.

The authors still need to correct minor errors in the manuscript. Please, check my comments below.

We look forward to receiving your revised manuscript.

Kind regards,

Maykon Passos Cristiano, D. Sc.

Academic Editor

PLOS ONE

Additional Editor Comments (if provided):

Please check the two files with the manuscript and the S3 table that I reviewed. In the introduction, I inserted the authorities and the years that the species was described (Page 3). Please, check if this information is correct. You should check the entire manuscript using this example and added the information when the first time the species or genus appears, and include the authority and year of description (Including in the title).

Please check my English corrections in table S3.

---

## [Author Response · Author response to Decision Letter 2]

16 Dec 2020

We reviewed the Editor's inquiries and uploaded a corrected version.

The corrections are highlighted in the text of the manuscript.

 Cordially,

Daniel Pacheco Bruschi

---

## [Editor Report · Decision Letter 3]

21 Dec 2020

PONE-D-20-23522R3

Cytogenetic characterization and mapping of the repetitive DNAs in Cycloramphus bolitoglossus (Werner, 1897): more clues for the chromosome evolution in the genus Cycloramphus (Anura, Cycloramphidae)

PLOS ONE

Dear Dr. Bruschi,

Thank you for submitting your manuscript to PLOS ONE. After careful consideration, we feel that it has merit but does not fully meet PLOS ONE’s publication criteria as it currently stands. Therefore, we invite you to submit a revised version of the manuscript that addresses the points raised during the review process.

There are some pending issues that need to be fixed before a final decision can be made.

We look forward to receiving your revised manuscript.

Kind regards,

Maykon Passos Cristiano, D. Sc.

Academic Editor

PLOS ONE

Additional Editor Comments (if provided):

There are still some species that do not have the authority who described them.

You should check the entire manuscript and added the information when the first time the species or genus appears, and include the authority and year of description (Including titles, captions and tables).

---

## [Author Response · Author response to Decision Letter 3]

21 Dec 2020

Our responses to the suggestions of the reviewers are presented in Response to Reviewer doc. We would like to thank all the reviewers for their important contributions for the improvement of the manuscript. The corrections are highlighted in the text of the manuscript.

 Cordially,

---

## [Editor Report · Decision Letter 4]

23 Dec 2020

Cytogenetic characterization and mapping of the repetitive DNAs in Cycloramphus bolitoglossus (Werner, 1897): more clues for the chromosome evolution in the genus Cycloramphus (Anura, Cycloramphidae)

PONE-D-20-23522R4

Dear Dr. Bruschi,

We’re pleased to inform you that your manuscript has been judged scientifically suitable for publication and will be formally accepted for publication once it meets all outstanding technical requirements.

Kind regards,

Maykon Passos Cristiano, D. Sc.

Academic Editor

PLOS ONE
---

## [Editor Report · Acceptance letter]

4 Jan 2021

PONE-D-20-23522R4 

Cytogenetic characterization and mapping of the repetitive DNAs in *Cycloramphus bolitoglossus* (Werner, 1897): more clues for the chromosome evolution in the genus *Cycloramphus* (Anura, Cycloramphidae) 

Dear Dr. Bruschi:

I'm pleased to inform you that your manuscript has been deemed suitable for publication in PLOS ONE. Congratulations! Your manuscript is now with our production department. 

Kind regards, 

on behalf of

Mr. Maykon Passos Cristiano 

Academic Editor

PLOS ONE